# Detectable Lipidomes and Metabolomes by Different Plasma Exosome Isolation Methods in Healthy Controls and Patients with Advanced Prostate and Lung Cancer

**DOI:** 10.3390/ijms24031830

**Published:** 2023-01-17

**Authors:** Alex C. Soupir, Yijun Tian, Paul A. Stewart, Yury O. Nunez-Lopez, Brandon J. Manley, Bruna Pellini, Amanda M. Bloomer, Jingsong Zhang, Qianxing Mo, Douglas C. Marchion, Min Liu, John M. Koomen, Erin M. Siegel, Liang Wang

**Affiliations:** 1Department of Tumor Biology, Moffitt Cancer Center, Tampa, FL 33612, USA; 2Department of Biostatistics and Bioinformatics, Moffitt Cancer Center, Tampa, FL 33612, USA; 3Advent Health, Translational Research Institute for Metabolism and Diabetes, Orlando, FL 32804, USA; 4Department of Genitourinary Oncology, Moffitt Cancer Center, Tampa, FL 33612, USA; 5Department of Thoracic Oncology, Moffitt Cancer Center, Tampa, FL 33612, USA; 6Department of Cancer Epidemiology, Moffitt Cancer Center, Tampa, FL 33612, USA; 7Tissue Core, Moffitt Cancer Center, Tampa, FL 33612, USA; 8Proteomics & Metabolomics Core, Moffitt Cancer Center, Tampa, FL 33612, USA; 9Department of Molecular Oncology, Moffitt Cancer Center, Tampa, FL 33612, USA

**Keywords:** exosome extraction, cancer multiome, plasma omics, circulating lipids, circulating metabolites

## Abstract

Circulating exosomes in the blood are promising tools for biomarker discovery in cancer. Due to their heterogeneity, different isolation methods may enrich distinct exosome cargos generating different omic profiles. In this study, we evaluated the effects of plasma exosome isolation methods on detectable multi-omic profiles in patients with non-small cell lung cancer (NSCLC), castration-resistant prostate cancer (CRPC), and healthy controls, and developed an algorithm to quantify exosome enrichment. Plasma exosomes were isolated from CRPC (n = 10), NSCLC (n = 14), and healthy controls (n = 10) using three different methods: size exclusion chromatography (SEC), lectin binding, and T-cell immunoglobulin domain and mucin domain-containing protein 4 (TIM4) binding. Molecular profiles were determined by mass spectrometry of extracted exosome fractions. Enrichment analysis of uniquely detected molecules was performed for each method with MetaboAnalyst. The exosome enrichment index (EEI) scores methods based on top differential molecules between patient groups. The lipidomic analysis detected 949 lipids using exosomes from SEC, followed by 246 from lectin binding and 226 from TIM4 binding. The detectable metabolites showed SEC identifying 191 while lectin binding and TIM4 binding identified 100 and 107, respectively. When comparing uniquely detected molecules, different methods showed preferential enrichment of different sets of molecules with SEC enriching the greatest diversity. Compared to controls, SEC identified 28 lipids showing significant difference in NSCLC, while only 1 metabolite in NSCLC and 5 metabolites in CRPC were considered statistically significant (FDR < 0.1). Neither lectin-binding- nor TIM4-binding-derived exosome lipids or metabolites demonstrated significant differences between patient groups. We observed the highest EEI from SEC in lipids (NSCLC: 871.33) which was also noted in metabolites. These results support that the size exclusion method of exosome extraction implemented by SBI captures more heterogeneous exosome populations. In contrast, lectin-binding and TIM4-binding methods bind surface glycans or phosphatidylserine moieties of the exosomes. Overall, these findings suggest that specific isolation methods select subpopulations which may significantly impact cancer biomarker discovery.

## 1. Introduction

Liquid biopsies have become a widely used as tool for molecular profiling, and more recently, for treatment response monitoring and minimal residual disease detection in many solid malignancies [1,2,3]. These platforms offer a minimally invasive way to perform comprehensive genomic testing without needing a tissue biopsy by using urine or blood as sources of tumoral DNA [4]. Many components of the blood or urine can be explored to measure disease burden, including cell-free DNA (cfDNA), cell-free RNA (cfRNA), circulating tumor cells (CTCs), exosomes, and other free-traveling molecules [5,6]. Exosomes are membrane vesicles (30–150 nm) consisting of a lipid bilayer membrane [7]. They are believed to be involved in various biological functions and may relay information between cells [8]. As exosome cargos contain cell-of-origin characteristics such as miRNA, proteins, lipids, and metabolites, exosomes are considered an ideal source for liquid biopsies using blood from cancer patients. Tumor-derived circulating exosomes may become a valuable source of tumor-specific molecules as they provide a protective environment for tumor-specific RNA, DNA, protein, lipids, and metabolites while enabling the capture of a tumor’s heterogeneity [6,9,10]. Additionally, recent studies have reported that patients with cancer have higher exosome numbers in plasma than healthy individuals [11,12].

Several methods have been employed to isolate exosomes from patient plasma samples, from original ultracentrifugation to size exclusion and affinity binding [7,13,14]. The simplest of these methods is size exclusion chromatography (SEC), which involves passing the sample through a column and collecting the flowthrough containing the exosomes for the downstream analyses [15]. In contrast, ultracentrifugation is more time consuming as it requires centrifugation at 100,000× *g* for several hours [15]. Other isolation methods take advantage of the exosome’s molecular characteristics to extract subpopulations. For example, lectins can be used to bind to the surface carbohydrates of exosomes producing aggregates that centrifugation can precipitate [7]. Similar to the lectin binding of surface markers, T-cell immunoglobulin domain and mucin domain-containing protein 4 (TIM4) have been used to bind to the phospholipid phosphatidylserine component of cell membranes and have been shown to increase the purity of exosome-specific molecules [13]. Since exosomes are extremely heterogeneous, one isolation method may enrich a specific subpopulation with cargo containing unique molecular contents. Therefore, the method used for exosome enrichment likely generates significant bias toward certain subsets of exosome populations.

Previously, it has been shown that cancer subtypes originated from cell lines produce different lipid signatures in their exosomes [16,17]. While metabolites have been less studied as biomarkers, the byproducts of cellular processes may provide distinct disease-related pathways of the exosomes’ tissue of origin, i.e., oncometabolites [18]. To date, it is unknown which exosome enrichment methodology is optimal in providing a comprehensive representation of tumor-derived exosomal lipids and metabolites. To evaluate commercially available isolation methods that may enrich cancer-related exosome cargos, we first performed mass spectrometry analysis to identify and quantify lipids and metabolites, respectively, in circulating exosomes from healthy individuals. We then performed the same analysis in circulating exosomes from patients with advanced non-small cell lung cancer (NSCLC) and castration-resistant prostate cancer (CRPC). Next, we developed an exosome enrichment index (EEI) from these results that identified differentially abundant lipids and metabolites in samples from patients with cancer compared to healthy controls.

## 2. Results

### 2.1. Various Exosome Concentrations and Size Distributions among Different Isolation Methods

NanoSight analysis was performed on pooled plasma from healthy donors to characterize the concentration and size distribution of exosomes isolated using different isolation methods. Exosome concentrations were measured across the three methods (two replicates), ranging from 2.1 × 10^9^ particles/mL of elution using TIM4 binding to 5.48 × 10^12^ particles/mL of elution using SEC. Figure 1A shows particle size and concentration from size exclusion chromatography. With 1:75 dilution, the SEC elution showed a single peak with a max concentration at 201 nm. Both lectin-binding and TIM4-binding methods produced single peak size distributions as well (Figure 1B,C, respectively). Particle concentration following lectin-binding exosome extraction was 1.03 × 10^10^ particles/mL, approximately a 5-fold higher concentration than TIM4-binding elutions. A summary of exosome analysis by NanoSight can be found in Table 1.

### 2.2. Batch Effects from Mass Spectrometry Data

Lipid and metabolite data were analyzed by PCA to identify any batch effects caused by different exosome extraction timepoints. The PCA plots of the lipid data showed the dimension reduction of the IRON-normalized log2 values for SEC, lectin binding, and TIM4 binding (Appendix A). Viewing the distribution across different diagnosis groups showed that the first principal component captured between 13.64% (SEC; Appendix A) and 11.39% (lectin binding; Appendix A) of the total variance. The second principal component captured between 8.96% (TIM4 binding; Appendix A) and 9.67% (SEC) of the total variance. There was no clear separation based on diagnosis groups with either of the principal components. We also examined the principal components to determine if there was any batch effect. We did not observe a separation on either the first or second principal component.

Metabolite profiles showed that sample 4 from the SEC method was an outlier by five standard deviations in the first principal component (Appendix A). Lectin binding and TIM4 binding did not produce any samples identified as an outlier (Appendix A). Additionally, this profile showed a slight separation of the two processing batches along the second principal component. As a result, ComBat was performed to adjust for the processing batches of all isolation methods after removal of sample 4 from the SEC dataset (Appendix A). The final PCA plots for patient groups (Appendix A) did not show any significant relationship between first and second principal component. Further, annotating for extraction batch shows extraction time was corrected following ComBat adjustment (Appendix A).

### 2.3. Significant Difference in Detectable Lipidome and Metabolome among Different Exosome Isolation Methods

Total lipid load from each exosome isolation method was investigated by comparing the number of uniquely identified molecules. From this analysis, we identified 949 unique lipid peaks covering all six lipid super classes (fatty acyls, glycerolipids, glycerophospholipids, prenol lipids, sphingolipids, and sterol lipids) in SEC samples. Total unique peaks with SEC were 3.9-fold higher than lectin binding (n = 246), and 4.2-fold higher than TIM4 binding (n = 226). Those lipids identified using the lectin-binding method enrichment spanned all six lipid super classes, however, elutions using the TIM4-binding method did not detect any lipids belonging to the prenol super class. Lipids identified from exosome fractions by SEC ranged from an average of 935 (standard deviation (SD) = 22) unique lipids in patients with NSCLC to an average of 943 (SD = 6) unique lipids in healthy donors’ samples (Figure 2A). The lectin-binding method detected an average of 245 (SD = 1) unique lipids in samples from controls and CRPC, and 244 (SD = 1) unique lipids from patients with NSCLC samples. However, the TIM4-binding method detected 204 (SD = 20), 208 (SD = 15), and 213 (SD = 11) in controls, NSCLC, and CRPC, respectively.

Similarly, total metabolite load from each exosome isolation method was explored by comparing the number of uniquely identified metabolite peaks. Metabolites that were identified by different isolation methods ranged from 100 using lectin binding to 191 total unique metabolites in at least 75% of samples using SEC missing that molecule (Figure 2B). The detectable metabolites with size exclusion ranged from an average of 188 (SD = 6) in NSCLC to 190 in CRPC (SD = 0) and controls (SD = 1). Lectin- and TIM4-binding elutions resulted in similar numbers of metabolites. TIM4-binding elutions had an average of 107 metabolites in NSCLC, PC, and controls while lectin-binding elutions had an average of 99 metabolites in NSCLC, PC, and healthy controls.

Since SEC detected significantly more lipids and metabolites than the other two methods, we wondered if the differences were caused by low abundance of these analytes. Using the lectin-binding method, we found that the mean value of lipids that overlapped with SEC was 17.88 (SD = 2.51, N = 175), which is significantly higher than the mean value (=16.85, SD = 2.65, N = 57) of lipids without overlapping (*p* = 0.011). Within the TIM4-binding method, we observed a similar trend with borderline significance (*p* = 0.058). The mean values of lipids that were overlapped and non-overlapped with the SEC method was 14.59 (SD = 2.24, N = 175) and 13.81 (SD = 2.75, N = 57), respectively. Clearly, undetected analytes in both lectin- and TIM4-binding elutions were partially attributable to their low abundance of the isolated exosomes. Appendix A shows the normalized intensity distribution of each lipid in three isolation methods.

### 2.4. Overlap of Detectable Lipidome and Metabolome among Different Exosome Isolation Methods

Unique molecules were compared between the isolation methods to identify those that were similar between and unique to each method. As the isolation methods were analyzed at different times, peaks were thus identified by compound names. Among the three methods, large lipid overlap was noted when comparing SEC with both lectin binding and TIM4 binding (Figure 3A). A total of 799 unique lipid names were identified with SEC, while 232 and 207 were identified using lectin binding and TIM4 binding, respectively, across all sample groups. Of the 799 lipids identified with SEC, 575 (72.0%) were unique to the SEC method, and 101 (12.6%) were shared by all three methods. Forty-two lipids unique to either lectin binding or TIM4 binding were identified in exosome fractions, accounting for 18.1% and 20.3% of total lipids identified with lectin- and TIM4-binding methods, respectively. The 101 common lipids shared across all methods represent 43.5% and 48.8% of the total lipids identified with lectin and TIM4 binding, respectively.

Unique metabolites across all samples (found in a minimum of three samples in each group) were matched between methods by metabolite name (Figure 3B). SEC identified 154 unique names, of which 46 were also shared with lectin binding and TIM4 binding. There were 87 unique metabolites identified from enriched exosomes with lectin binding and 94 with TIM4 binding. SEC contained the largest proportion of unique metabolites, at 42.9%, of 154. The 46 shared metabolites represented 29.9%, 52.9%, and 48.9% of total detectable metabolites in SEC, lectin binding, and TIM4 binding, respectively.

### 2.5. Enrichment of Unique Lipids and Metabolites from Each Exosome Isolation Method

We applied MetaboAnalyst (23 August 2022) to perform metabolite set enrichment analysis using unique lipids and metabolites from each isolation method (Figure 4). In chemical structures, unique lipids from SEC showed significant enrichment in the super classes of glycerophospholipids (FDR = 3.11 × 10^−15^) and sphingolipids (FDR = 4.45 × 10^−9^). Main class analysis revealed six significantly enriched lipid sets: glycerophophocholines, ceramides, sphingomyelins, glycerophosphoethanolamines, diradylglycerols, and glycerophosphoinositols (FDR = 8.74 × 10^−26^, 1.11 × 10^−22^, 2.52 × 10^−21^, 3.16 × 10^−6^, 3.39 × 10^−4^, and 3.21 × 10^−3^, respectively). Subclass analysis identified 14 significant lipid sets, of which diacylglycerophosphocholines were the most significant (FDR = 2.99 × 10^−42^). The unique lipids identified with lectin binding showed a significantly enriched super class of glycerolipids (FDR = 0.037), and subclasses of triacylglycerols (FDR = 0.0001) and diacylglycerols (FDR = 0.0036). The unique lipids from TIM4 binding did not show significant enrichment in super class and main class. However, subclass analysis did identify significant enrichments in triacylglycerols (FDR = 1.72 × 10^−6^) and diacylglycerophosphoserines (FDR = 0.0553). Appendix A shows detailed enrichment analysis.

For metabolites, the unique molecules from the SEC method showed significant enrichment in chemical structure of the super classes of organic acids (FDR = 6.13 × 10^−27^), carbohydrates (FDR = 2.45 × 10^−17^), benzenoids (FDR = 6.11 × 10^−14^), fatty acyls (FDR = 3.76 × 10^−5^), nucleic acids (FDR = 0.012), and organoheterocyclic compounds (FDR = 0.056; Figure 5). Main class analysis revealed significant enrichment in 18 sets with phenylacetic acids as the most significantly enriched (FDR = 4.99 × 10^−9^). Subclass analysis identified amino acids as the most significantly enriched set (FDR = 9.44 × 10^−11^). The unique metabolites from lectin binding showed significant enrichment in super classes of organic acids (FDR = 1.58 × 10^−5^), nucleic acids (FDR = 1.17 × 10^−3^), benzenoids (FDR = 3.22 × 10^−3^), organoheterocyclic compounds (FDR = 3.87 × 10^−3^), and carbohydrates (FDR = 0.055). Main class analysis identified six significantly enriched metabolite sets with pyridoxines as the most significant (FDR = 0.055). Subclass analysis showed an enrichment of dicarboxylic acids (FDR = 0.038) and pyridoxines (FDR = 0.0553). The unique metabolites identified using the TIM4-binding method showed significant enrichment of amino sugar and nucleotide sugar metabolism (FDR = 0.0296). Super class analysis showed significant metabolite sets of nucleic acids (FDR = 3.18 × 10^−7^), carbohydrates (FDR = 2.18 × 10^−3^), organic acids (FDR= 2.36 × 10^−3^), and benzenoids (FDR = 0.0143). Main class analysis revealed significant enrichment in purines (FDR = 6.17 × 10^−5^), monosaccharides (FDR = 0.01), amino acids and peptides (FDR = 0.0107), and imidazoles (0.0995). Subclass analysis produced significant enrichment in xanthines (FDR = 1.34 × 10^−6^) and amino acids (FDR = 2.81 × 10^−3^).

### 2.6. Differential Lipidomes and Metabolomes between Different Exosome Isolation Methods in Different Cancer Types

Differences between patients with NSCLC and CRPC and healthy donors were identified from IRON-normalized and log_2_ abundances using Student’s *t*-tests. When compared to the control group, significant differences were found in lipids from SEC elutions in NSCLC samples (Appendix A). A total of 28 lipids were significant at FDR < 0.1, with 26 showing higher abundance, and another 2 showing lower abundance in samples from patients with NSCLC. This differential abundance of lipids was not observed in elutions from lectin or TIM4 binding (Appendix A). Additionally, when comparing CRPC samples to the control samples, no difference in lipid abundance was identified (Appendix A).

The metabolites identified in exosome fractions from SEC only showed a single metabolite as being significantly different between NSCLC and control samples (Appendix A). Neither lectin nor TIM4 binding produced exosome fractions with significant differences in metabolites at FDR < 0.1 between NSCLC and controls (Appendix A). In contrast to the single metabolite that was different between NSCLC and controls, SEC identified four significantly higher and one significantly lower metabolite when comparing CRPC to controls (Appendix A). The lectin- and TIM4-binding methods again failed to identify any of these differences (Appendix A).

We also performed clustering analysis using the 25 most significant lipids and metabolites to cluster patients (Table 2 and Table 3). First, we used lipids and metabolites separately to cluster patients (single-omic approach, Appendix A). Then, we clustered patients using lipids and metabolites together (multi-omic). For the multi-omic clustering approach, patients were clustered by their profile while the molecules were clustered within their respective classes. For SEC data, although not all significantly different between NSCLC samples and controls or CRPC samples and controls, the top 25 most significant molecules for lipids and metabolites were able to cluster most samples into the patient groups respectably (Figure 6A,B), with the exception of three NSCLC patients clustering within the controls and one CRPC that did not cluster with the rest of the patient group. With lectin-binding data, six NSCLC samples clustered within control samples while four CRPC samples were clustered with the controls (Appendix A). The multi-omic clustering using TIM4-binding data clustered two NSCLC samples and one CRPC PC sample in controls (Appendix A).

### 2.7. Enrichment of Cancer-Related Exosomes in Patients with Cancer

To quantify the enrichment of cancer-related exosomes, we developed an EEI using the top 50 most significant molecules by adjusted Student’s *t*-test *p*-values, giving positive values to those that appeared in more cancer samples and negative values to those that appeared in more healthy controls. The EEI for lipids showed which sample group and how much or how significant it was being enriched in a particular method (Table 4). SEC showed the highest EEIs for NSCLC and CRPC compared to the controls. Specifically, the EEI (=871.33) from NSCLC with SEC was the highest among all groups. This EEI was consistent with the trend that the SEC method captured more lipids and showed the most significant differentially abundant lipids in patients with NSCLC over control samples. Interestingly, all isolation methods produced positive EEIs (>0), suggesting that even when there were few differential lipids with the lectin- and TIM4-binding methods, there was still clearly a trend that lipids with greater abundance showed up more frequently in the patients with cancer than in healthy donors. Figure 7A shows the positive EEI trend for all three methods when comparing samples from patients with cancer to control samples.

Further investigation of the most significant metabolites showed the highest EEI from SEC for both NSCLC and CRPC (Table 4). While the EEI for metabolites was not as large as the one for lipids, the EEI from the SBI method for CRPC was 571.77, which was still 2.4-fold and 2.0-fold higher than the lectin-binding (231.55) and TIM4-binding (290.65) EEI, respectively. The metabolites listed in Table 3 contribute to the EEI. When compared to CRPC, NSCLC showed lower EEI using the three methods, though SEC still showed the largest positive EEI (=347.37) in the patient group, indicating enrichment of the cancer-related metabolites in the exosome elution fractions. This result is depicted in Figure 7B where all peaks for both NSCLC and CRPC for the three methods with positive values are large.

## 3. Discussion

In this study, we set out to identify which commercially available plasma exosome isolation kit, which implemented either size exclusion chromatography, lectin binding, or TIM4 binding as a means of capturing exosomes, provided the most information that could be used in comparing patient groups. To do this, we compared the lipidome and metabolome of plasma exosomes isolated with these three different isolation methods. We demonstrated that enriched fractions produced by the lectin-based and TIM4-based exosome isolation methods produced a relatively concentrated particle size. In contrast, distributions from size exclusion chromatography produced exosomes with a relatively dispersed size range. We also demonstrated that the size exclusion method detected the largest number of lipids and metabolites (including different peaks of the same lipid ID) through LC-MS/MS and LC-HRMS analysis, as well as the highest number of uniquely named molecules across the three methods examined. Furthermore, we demonstrated that the size exclusion method produced the largest exosome enrichment index (EEI), indicating that enriched fractions from this method retain more abundant exosomes from cancer patients (both NSCLC and CRPC) than healthy controls. These data suggest that SEC provides the greatest spectrum of lipids and metabolites from which to further identify cancer-related exosome cargo. However, this method only enriches for exosome size and therefore isolates a broad spectrum of circulating exosomes, most of which may not represent tumor-derived exosomes but rather exosomes with molecules that have higher abundance in patients with cancer. Therefore, sample size considerations and analytical methods should consider a possible high false positive rate.

Multi-omic approaches capture more comprehensive molecular snapshots than conventional single-omic studies in cardiovascular disease and cancers [19,20,21,22]. In this study, we showed that different methods enrich exosome subpopulations with various lipids and metabolites. By using lipidome and metabolome together, our classification was significantly improved compared to the single omics for hierarchical clustering. This demonstrates the benefit of the multi-omic approach for clustering NSCLC and CRPC with healthy controls over lipidome or metabolome separately.

While our data demonstrate a large overlap between the lipidome and metabolome identified between the three methods, there were many molecules (especially identified from the SEC exosome fraction) that were uniquely identified with each method. Since SEC does not discriminate subpopulations based on exosomal surface markers, many of the SEC-identified lipids were not present in exosomes that readily bind to a lectin or TIM4. One possibility is that SEC isolates a larger number of exosomes and other extracellular vesicles as well as non-vesicular particles. To assess the potential contamination of non-vesicular components, detergent lysis is commonly used. However, this has also been shown to disrupt lipoproteins that would make assessing contaminants more difficult [23]. Proteinase K could also be used though integrity of extracellular vesicles may be compromised, causing confounding results. Alternatively, the lower levels of recovery using specific exosome enrichment binding with lectin and TIM4 could lead to loss of detection for low-abundance analytes. These differences highlight that lectin and TIM4 binding may enrich exosome subpopulations that are specific to the method used. Biomarkers in diseases involved in tissue cells with highly abundant lectin or TIM4 may benefit from these methods. A previous study has described subpopulations of exosomes extracted from cancer cell lines [24]. Detail multi-omic analysis of tumor-derived exosomes may help identify target molecules for separation of exosome cargos that are enriched in patients with cancer.

Enrichment analysis showed distinct molecular profiles in unique lipids and metabolites from each of the three isolation methods. The differentially enriched molecular profiles could be attributed to the exosome isolation method. Using the SEC method, the isolated exosomes should contain a full spectrum of exosome cargos, which is consistent with our observation that the SEC method not only generated the highest exosome yields but also captured more diverse categories of lipids and metabolites. Meanwhile, the lectin- and TIM4-binding methods are expected to capture exosome subpopulations with less diverse molecule distribution, which is also supported by our observation that these methods produced smaller numbers of exosomes and fewer enriched molecule sets. Clearly, selection of the exosome isolation method is critical for molecular biomarker study.

This study was able to identify significant differences in exosome lipid and metabolic profiles in plasma exosomes isolated using SEC between CRPC and NSCLC and healthy controls. Metabolite analysis revealed that L-cystathionine was higher in CRPC samples than in samples from patients with NSCLC or healthy individuals (Table 3). This finding is in agreement with data from a study of prostate cancer cell line PC-3 derived from bone metastasis that showed cystathionine γ-lyase is upregulated in these cells [25]. In addition, we found that the abundance of metabolite N^a^-acetyl-L-lysine was significantly lower in the NSCLC samples compared to healthy donors’ samples. It has been shown that acetylation of lysine residues is an important regulator mechanism in epigenetics controlling DNA binding to histones and maintaining stability [26]. Deletions and mutations of acetyltransferase genes have been reported in several cancers including lung cancer while overexpression has been reported for others, such as prostate cancer [27]. Further investigation of the contribution of lower N^a^-acetyl-L-lysine in plasma exosomes of lung cancer patients could help identify its relationship to exosomes and cancer.

While all methods produced positive EEIs, indicating that there was enrichment of higher abundance molecules isolated in samples from cancer patients compared to controls, none of these methods used an approach for exosome extraction that would enrich exosomes using known cancer-specific biomarkers. For example, the prostate-cancer-specific biomarker *FOLH1* has been found on prostate-derived exosomes [28] and EGFR protein has been validated as an exosomal component for NSCLC cell lines with high expression of EGFR [12,29,30]. An antibody against these proteins may be used to enrich prostate- or lung-cancer-specific exosomes. However, there is still limited understanding of the full spectrum of tumor-specific exosome markers. Using a broader isolation approach may be important to identify unknown cancer-specific exosome biomarkers that may not be captured due to exosome heterogeneity [15].

Our study has demonstrated the potential utility of exosome enrichment techniques to identify lipids and metabolites discriminative between patients with cancers and controls; however, it does have a few limitations to consider. First, this study only tested three commercially available methods, each enriching different exosome subpopulations. This work should be extended to test cancer-specific molecules in exosomes (such as the use of EpiCan or PanCK antibody). Second, this multi-omic study has a modest sample size with a total of 34 participants. While we used stringent criteria for patient enrollment and frequency matching of healthy controls, a larger patient cohort is needed to validate these findings. In addition, we were only able to include two omic platforms in this study. Future work should expand to include recruitment of more diverse cancer types and larger patient cohorts, and additional omics, such as proteomics and/or transcriptomics from exosomes. From the increased patient diversity and captured omics, a targeted panel could be created with those molecules most different between patient groups. Lastly, our study only evaluated two cancer types (NSCLC and CRPC). Inclusion of other cancers in future study may provide more distinct pan-cancer multi-omic fingerprints in plasma exosomes.

## 4. Materials and Methods

### 4.1. Plasma Collection and Storage

All participants were submitted to the Total Cancer Care Protocol, Moffitt Cancer Center’s institutional biorepository (MCC#14690; Advarra IRB Pro00014441, Table 5). Blood volumes from 5 to 10 mL were collected in EDTA K2 vacutainers from patients with CRPC (n = 10) and stage IV NSCLC (n = 14), along with healthy individuals (n = 10) frequency matched on age and gender (blood draw information is included at Appendix A). Advanced disease was selected to increase likelihood of capturing cancer-specific exosomes due to high tumor burden. Patient samples were collected prior to systemic treatment start or after a washout period of two weeks when switching treatment. Plasma samples were centrifuged within 45 min (min) of collection at 600× *g* for 10 min at room temperature (RT) after which plasma was transferred to a new 10 mL centrifuge tube. Plasma fraction was centrifuged again at 9000× *g* for 10 min at RT to produce platelet-poor plasma. Aliquots of 500 µL plasma were immediately stored at −80 °C until exosome processing.

### 4.2. Plasma Exosome Extraction

Exosome extraction kits used in this study included (1) SBI SmartSEC™ Single for EV Isolation™ (Palo Alto, CA, USA; cat# SSEC200A-1), (2) Takara Capturem™ Extracellular Vesicle Isolation Kit (Mini) (Kusatsu, Shiga, Japan; cat# 635741), and (3) Fujifilm Wako MagCapture™ Exosome Isolation Kit PS (Chuo-Ku, Osaka, Japan; cat# 295-74001). These kits use different methods for exosome isolation from patient plasma. SBI SmartSEC™ employs size exclusion chromatography (SEC), Takara Capturem™ uses columns containing lectin to select exosomes (lectin binding), and Wako MagCapture™ relies on T-cell immunoglobulin domain and mucin domain-containing protein 4 (TIM4) to selectively bind to exosomes (TIM4 binding).

Preparation for all kits included thawing plasma aliquots (total of 1250 µL per patient) at RT, transferring to 1.5 mL microcentrifuge tubes, and centrifuging at 10,000× *g* for 15 min at 4 °C to remove large vesicles and cellular debris. The supernatant was then used for exosome extraction. Manufacturer protocols were used for all three methods. SEC required 250 µL of plasma for exosome extraction, while lectin binding and TIM4 binding required 500 µL of plasma. The SEC method followed the manufacturer’s manual. Samples for the lectin-binding method were centrifuged through a 100 kDa Amicon filter (Millipore, Burlington, MA, USA) and the retentate (volume remaining above the filter) was used for exosome isolation. For the TIM4-binding method, two binding–elution cycles were performed. Plasma samples were processed in two random batches: the first with 14 samples and the second with 20 samples. Exosome elutions from each method were divided into two fractions for lipidomics and metabolomics. The project workflow is depicted in Figure 8.

### 4.3. NanoSight Characterization of Exosome Size and Concentration

To characterize the size of particles extracted using the different technologies of SEC, lectin binding, and TIM4 binding, we implemented NanoSight NS300 with NTA Version 3.4 (build 3.4.003). Two samples of healthy pooled plasma (Innovative Research, Clearwater, FL, USA) were extracted with the above volumes for each kit. Samples from lectin binding were desalted with a 10 kDa Amicon filter, and samples from TIM4 binding were passed through a 0.45 µm filter to remove beads. Eluted samples were diluted with cell culture grade water by vortexing them for 30 s to suspend exosomes well, then loaded into the machine. For each sample, five captures were collected of 60 s each and all runs were used to determine sample concentration and particle size.

### 4.4. Mass Spectrometry Sample Preparation

Lipid fractions (200 µL, 40 µL, and 70 µL for SEC, lectin-binding, and TIM4-binding elutions, respectively) were thawed and mixed with 7.5 µL of 5× diluted SPLASH Lipidomix Standard. Chloroform and methanol (600 µL and 1.5 mL, respectively, for SBI; 120 µL and 300 µL, respectively, for lectin binding; 210 µL and 525 µL, respectively, for TIM4 binding) were added, vortexed vigorously, and then centrifuged at 2000× *g* for 10 min at RT to separate phases. The lower phase containing lipids was removed and the upper phase was washed with chloroform three times, pooling each wash. Pooled lipids were dried under 5 psi nitrogen at RT followed by suspension in methanol. Particulates were precipitated with centrifugation at 2000× *g* for 5 min at 5 °C and the supernatants were transferred to LC-MS autosampler vials.

All metabolite preparations were performed on ice. Metabolite fractions (200 µL, 40 µL, and 70 µL for SEC, lectin-binding, and TIM4-binding elutions, respectively) were thawed and mixed with precooled 80% methanol (800 µL, 160 µL, and 280 µL for SEC, lectin-binding, and TIM4-binding elutions, respectively) added to precipitate proteins and stored at −80 °C for 1 h, vortexed, and centrifuged at 18,800× *g* for 10 min at 0 °C. Samples were then incubated at −80 °C for 30 min followed by another centrifugation at 18,800× *g* for 10 min at 4 °C. Each metabolite supernatant was transferred to a new tube, dried, and redissolved in 20 µL 80% methanol before being transferred to autosampler vials.

### 4.5. Mass Spectrometry Analysis

Liquid chromatography with tandem mass spectrometry (LC-MS/MS) was used for lipid analysis on a Vanquish UPLC interfaced with a Q Exactive HF hybrid quadrupole-orbital ion trap mass spectrometer. Chromatographic separation was performed on a Brownlee SPCC C18 column (2.1 mm ID × 75 mm length, 2.7 µm particle size, Perkin Elmer Waltham, MA, USA) using mobile phases A (100% H_2_O containing 0.1% formic acid and 1% of 1 M NH_4_OAc) and B (1:1 acetonitrile: isopropanol containing 0.1% formic acid and 1% of 1M NH_4_OAc). The gradient was programmed as follows: 0–2 min 35% B, 2–8 min from 35% B to 80% B, 8–22 min from 80% B to 99% B, 22–36 min 99% B, 36.1–40 min from 99% to 35% B, with a flow rate of 0.400 mL/min. Lipid profile was monitored using data-dependent tandem mass spectrometry for the 10 highest intensity ion signals (5 positive and 5 negative in separate experiments). The MS instrument settings were as follows: sheath gas 50, auxiliary gas 10, sweep gas 1, spray voltage 3.5 kV, capillary temperature 325 °C, S-lens RF level 30, and the auxiliary gas heated at 350 °C, the scan range from *m*/*z* 120–1000, resolution 120,000 for MS and 30,000 for MS/MS, AGC target 3E6 for full MS and 1E5 for MS^2^, allowing ions to accumulate for up to 200 ms for MS and 50 ms for MS/MS. For MS/MS, the following settings were used: isolation window width 1.2 *m*/*z*, stepped NCE at 20, 30, and 40 a.u., minimum AGC 5E2, and dynamic exclusion of previously sampled peaks for 8 s. LipidSearch 4.2.2.1 (Thermo) was used for the identification of lipids.

Metabolite analysis was performed with ultra-high-performance liquid chromatography–high-resolution mass spectrometry (UHPLC-HRMS) using a Vanquish UHPLC interfaced with a Q Exactive HF quadrupole–orbital ion trap mass spectrometer (Thermo, San Jose, CA, USA). Chromatographic separation was performed using a SeQuant ZIC-pHILIC guard column (2.1 mm ID × 20 mm length, 5 µm particle size) and a SeQuant ZIC-pHILIC LC column (2.1 mm ID × 150 mm length, 5 µm particle size, MilliporeSigma, Burlington, MA, USA). Mobile phase A was aqueous 10 mM ammonium carbonate and 0.05% ammonium hydroxide, and mobile phase B was 100% acetonitrile. The gradient program included the following steps: start at 80% B, a linear gradient from 80 to 20% B over 13 min, stay at 20% B for 2 min, return to 80% B in 0.1 min, and re-equilibration for 4.9 min for a total run time of 20 min, and a flow rate was set to 0.400 mL/min. The autosampler was cooled to 5 °C and the column temperature was set to 30 °C. Sample injection volume was 2 µL for both positive ion mode and negative ion mode electrospray ionization. Full MS was performed in positive and negative mode separately detecting ions from *m*/*z* 65 to *m*/*z* 900. In addition, data-dependent acquisition was used for tandem mass spectrometry (MS/MS) of analytes in the pooled samples to enable verification of selected metabolites to confirm assignments. MZmine (v3.53) was used to identify and quantify metabolites by matching by *m*/*z* and RT to an in-house library.

### 4.6. Normalization and Principal Component Analysis for Outliers

LC-MS/MS data normalization was performed with iterative rank-order normalization (IRON) [31] for lipidomics and metabolomics data. Normalized data were analyzed with R v4.1.1 (CRAN) removing omic data that appeared in less than 75% of samples for lipids and metabolites (Appendix A), and outlier samples were identified via principal component analysis (PCA). If the sample was greater than 5 standard deviations from the mean, the whole sample was removed. Annotated metabolites were batch corrected with ComBat [32], after outlier removal, to adjust for the influence of extraction date. Abundance comparisons between patient groups were performed using Student’s *t*-test (unequal variance). For analytes with missing values, Fisher’s exact test was used to determine differences in frequency between patients with cancer and healthy controls. FDR < 0.1 was considered statistically significant (Appendix A). In addition to comparisons of lipids between patient groups, comparisons that identified more than 25 significantly different molecules were also analyzed with BioPan to find most active reaction chains and genes involved [33,34]. The default parameters were used with a *p*-value threshold of 0.05.

### 4.7. Overlap between Different Exosome Isolation Methods

To determine overlap between SEC, lectin binding, and TIM4 binding, molecules were filtered to be included in at least three samples to encompass as many molecules as possible while removing possible false identifications. Names of the molecules (disregarding retention time, CalcMz, and IonFormula) were passed to BioVenn (v1.1.3) to create Venn diagrams of the three methods. To create the Venn diagram for lipids and metabolites, unique names were used for the identified molecules. Since each kit was run separately, slight differences were noted in retention time and mass-to-charge ratios between batches. Of the unique molecules that were identified between the kits with BioVenn, we used MetaboAnalyst (metaboanalyst.ca) to carry out enrichment analysis of both lipids and metabolites to determine if a kit enriched a specific compound structure of a molecule.

### 4.8. Exosome Enrichment Index

For constructing an exosome enrichment index (EEI), we selected the smaller *p*-value between Fisher’s exact test and Student’s *t*-test and log_2_ transformed the data. If samples from healthy donors were enriched for a specific molecule (i.e., 4 of 10 CRPC and 9 of 10 healthy controls), a higher level of the molecule was considered negatively enriched in cancer cells and multiplied by −1. The top 50 most significant molecules were then summed creating an enrichment score that takes both significance and proportion of samples into consideration: a higher EEI indicates more significant enrichment of cancer-related molecules while negative EEI indicates more significant enrichment of molecules specific to control samples. Values were normalized to 100 molecules to account for the possibility of a low number of molecules being identified.

## 5. Conclusions

Our study evaluated the lipidome and metabolome profiling of plasma exosomes enriched with three commercially available exosome extraction methods. We showed that SBI SmartSEC™ which implements size exclusion chromatography produced the largest number of lipids and metabolites, as well as the largest EEI, indicating its ability to identify the most cancer-related lipids and metabolites differentiating between patient groups and controls. In addition to the lipidomics and metabolomics conducted in this study, we foresee that miRNA and the proteome could also provide a greater picture of the whole “circulome” profile of cancer-derived exosomes. Based on our findings, future studies using the SEC isolation method are preferred to identify cancer-derived biomarkers for early cancer detection and outcome prediction.

## Figures and Tables

**Figure 1 ijms-24-01830-f001:**
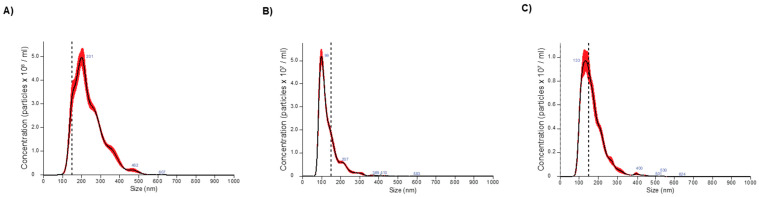
Size distribution of exosomes extracted using 3 different commercial kits. (**A**) SEC with peak size of 201 nm. (**B**) Lectin binding with peak size of 99 nm. (**C**) TIM4 binding with peak size of 133 nm.

**Figure 2 ijms-24-01830-f002:**
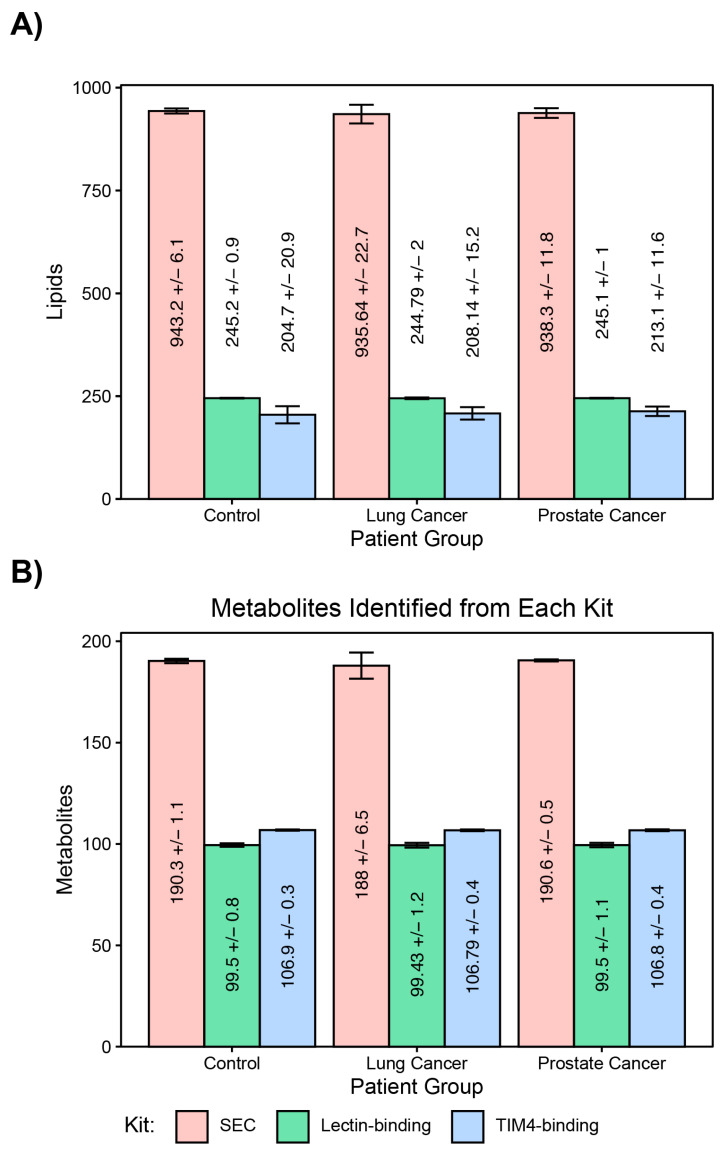
Average number of uniquely identified lipid (**A**) and metabolite (**B**) peaks from 3 exosome extraction methods.

**Figure 3 ijms-24-01830-f003:**
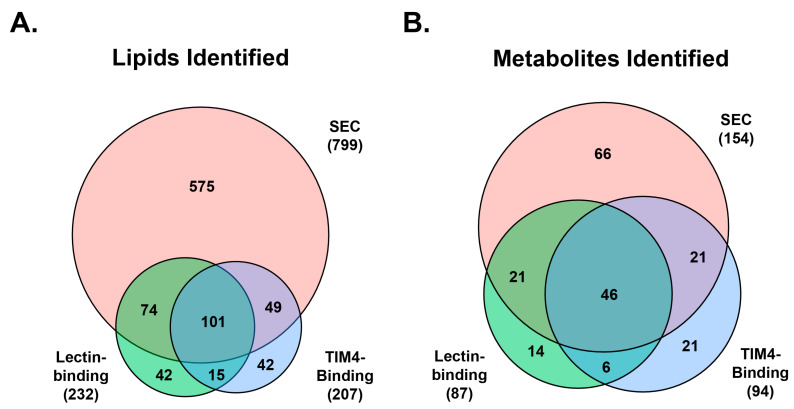
Overlap in identified lipids (**A**) and metabolites (**B**) for the 3 exosome extraction methods.

**Figure 4 ijms-24-01830-f004:**
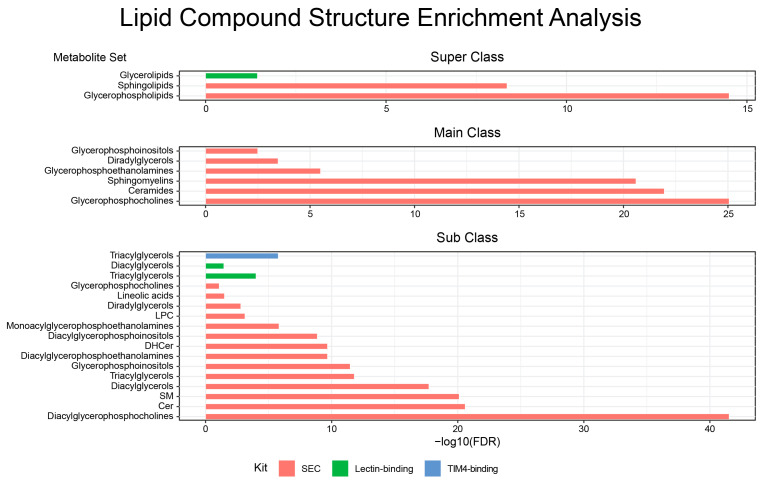
Enrichment of compound classes for each exosome isolation method using unique lipids. Results shown are FDR < 0.1.

**Figure 5 ijms-24-01830-f005:**
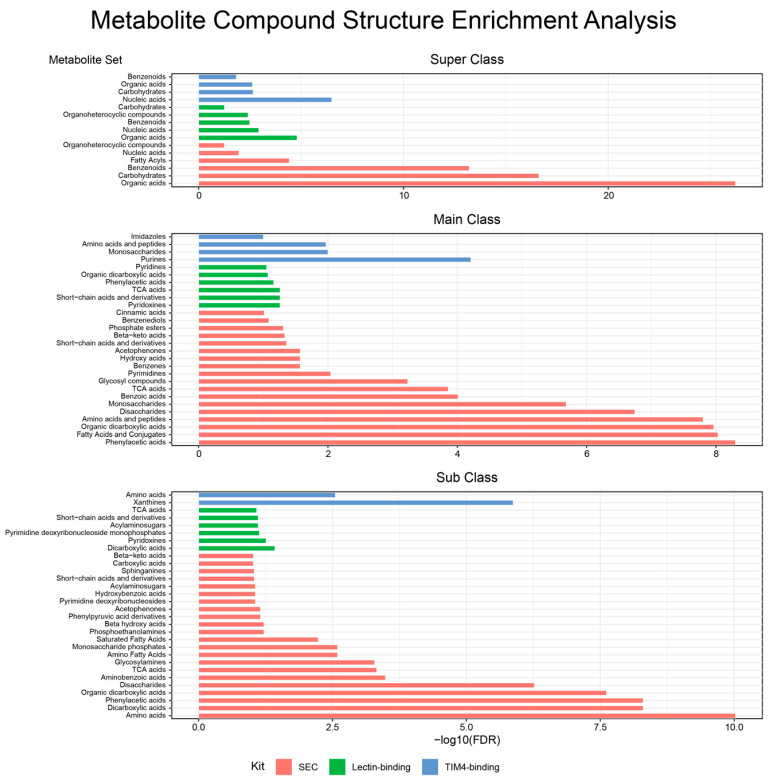
Enrichment results for compound classes with unique metabolites that each exosome isolation method produced. Results pass FDR < 0.1.

**Figure 6 ijms-24-01830-f006:**
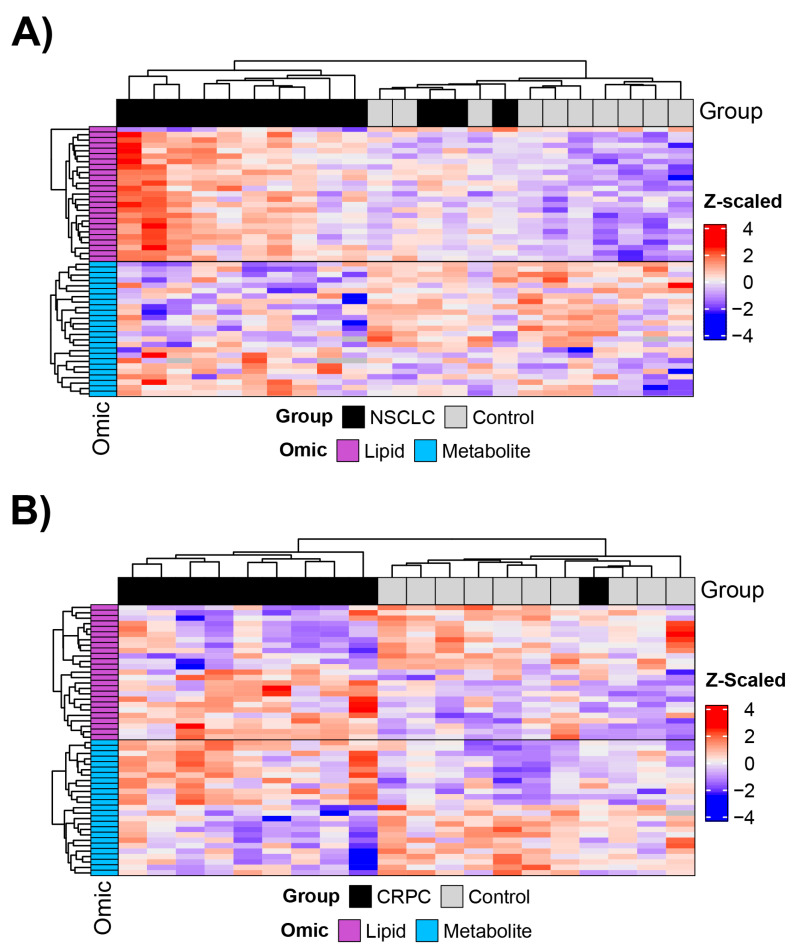
Heatmaps of the top 25 lipids and metabolites identified by SBI method between (**A**) lung cancer samples and healthy controls and (**B**) prostate cancer samples and healthy controls.

**Figure 7 ijms-24-01830-f007:**
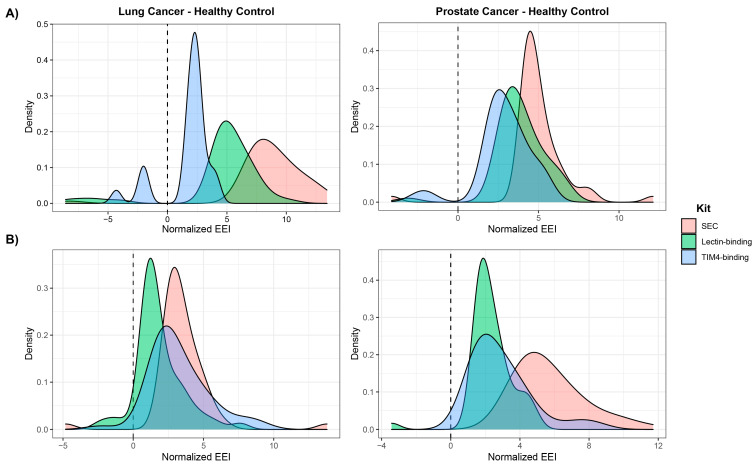
Exosome enrichment index (EEI) produced by the top 50 significant molecules of (**A**) lipids and (**B**) metabolites when comparing cancer samples to healthy controls. Size exclusion chromatography shows consistently higher EEI than the other two methods.

**Figure 8 ijms-24-01830-f008:**
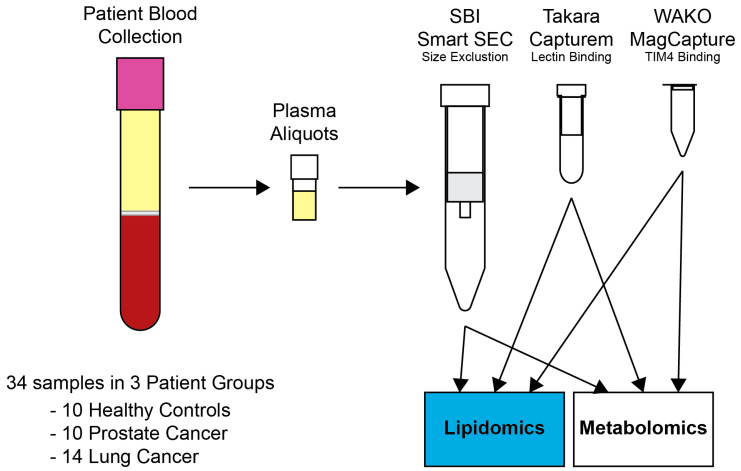
Outline of sample collection, storage, exosome extraction, and omic analysis.

**Table 1 ijms-24-01830-t001:** NanoSight results for 2 replicates of each exosome extraction method.

Kit	Sample	Dilution	Particles per Frame	SE	Centers per Frame	SE	Concentration (*p*/mL Dilution)	SE	Original Concentration (p/mL Elution)	Average Original Concentration (p/mL Elution)
SEC	Control 1	1/150	11.1	1.1	14.3	1.5	1.03 × 10^10^	1.00 × 10^9^	1.55 × 10^12^	3.51 × 10^12^
	Control 2	1/75	165.6	7.6	180.2	5.7	7.30 × 10^10^	3.40 × 10^9^	5.48 × 10^12^	
Lectin binding	Control 1	1/4	61.8	2	62.5	1.8	1.81 × 10^9^	8.10 × 10^7^	7.24 × 10^9^	1.03 × 10^10^
	Control 2	1/4	125	4.6	118.9	3.9	3.35 × 10^9^	1.19 × 10^8^	1.34 × 10^10^	
TIM4 binding	Control 1	1/2	77.2	1.3	75.5	1.2	1.05 × 10^9^	2.24 × 10^7^	2.10 × 10^9^	2.23 × 10^9^
	Control 2	1/2	87	3.8	85.3	3.3	1.18 × 10^9^	5.18 × 10^7^	2.36 × 10^9^	

**Table 2 ijms-24-01830-t002:** Top 5 differentially abundant lipids from SEC enriched exosomes in NSCLC and CRPC.

	Lipid Ion	Cancer Mean	Control Mean	Cancer Samples	Control Samples	*p*-Value	FDR
NSCLC vs. Controls	Cer(d18:2/22:0)+HCOO	21.61	20.97	14	10	0.0001	0.038
	Cer(d18:1/24:1)+HCOO	16.13	15.28	14	10	0.0002	0.038
	PC(15:0/18:3)+H	25.30	24.72	14	10	0.0002	0.038
	SM(d18:2/20:3)+H	21.88	21.02	14	10	0.0001	0.038
	SM(t18:1/18:1)+H	17.79	16.58	14	10	0.0002	0.038
CRPC vs. Controls	Cer(d18:2/23:0)+HCOO	17.50	18.18	10	10	0.0002	0.214
	Cer(d18:1/20:0)+HCOO	19.10	18.16	10	10	0.0033	0.997
	PE(16:1e/22:6)-H	18.83	19.64	10	10	0.0037	0.997
	PI(18:1/18:2)-H	14.69	13.70	10	10	0.0056	0.997
	TG(18:0/18:1/22:5)+NH4	18.44	19.88	10	10	0.0099	0.997

**Table 3 ijms-24-01830-t003:** Top 5 differentially abundant metabolites from SEC enriched exosomes in NSCLC and CRPC.

	Identity Mapped	Cancer Mean	Control Mean	Cancer Samples	Control Samples	*p*-Value	FDR
NSCLC vs. Controls	(+) Nalpha-Acetyl-L-Lysine	17.26	18.49	13	10	7.03 × 10^−5^	0.013
	(+) L-Proline	29.61	29.95	13	10	1.19 × 10^−2^	0.731
	(+) L-Lysine	25.60	25.88	13	10	0.0201	0.731
	(+) O-Acetyl-L-carnitine	28.31	27.66	13	10	0.0255	0.731
	(−) L-Proline	22.00	22.51	13	10	0.0301	0.731
PC vs. Controls	(+) L-Cystathionine	16.47	14.97	10	10	0.0003	0.057
	(−) L-Cystathionine	16.03	14.37	10	10	0.0009	0.062
	(−) 4-Acetamidobutanoic acid	18.54	17.47	10	10	0.0012	0.062
	(−) N-Acetyl-L-Alanine	20.65	19.95	10	10	0.0015	0.062
	(−) Lauric acid	23.58	24.41	10	10	0.0016	0.062

**Table 4 ijms-24-01830-t004:** Exosome enrichment index (EEI) indicating that SEC identified the most significant molecules between patients and controls.

Omic	Kit	Comparison	Raw	Molecules	Normalized	Up	Down
					EEI		
Lipids	SEC	NSCLC—Control	435.66	50	**871.33**	26	2
		CRPC—Control	248.92	50	**497.83**	0	0
	Lectin binding	NSCLC—Control	229.83	50	459.67	0	0
		CRPC—Control	193.09	50	386.19	0	0
	TIM4 binding	NSCLC—Control	83.47	50	166.94	0	0
		CRPC—Control	142.97	50	285.93	0	0
Metabolites	SEC	NSCLC—Control	173.69	50	**347.37**	0	1
		CRPC—Control	285.88	50	**571.77**	4	1
	Lectin binding	NSCLC—Control	91.09	50	182.19	0	0
		CRPC—Control	115.77	50	231.55	0	0
	TIM4 binding	NSCLC—Control	160.91	50	321.81	0	0
		CRPC—Control	145.32	50	290.65	0	0

**Table 5 ijms-24-01830-t005:** Clinical characteristics of patients and healthy controls.

Sample No.	Group	Gender	Current Age	Age at Diagnosis	Stage at Diagnosis	Treatment Status (Prior to Blood Collection)	Time from Last Treatment to Blood Collection
36	Control	F	69	NA	NA	NA	NA
26	Control	F	64	NA	NA	NA	NA
39	Control	M	72	NA	NA	NA	NA
37	Control	F	69	NA	NA	NA	NA
38	Control	F	65	NA	NA	NA	NA
40	Control	M	73	NA	NA	NA	NA
34	Control	M	69	NA	NA	NA	NA
41	Control	F	72	NA	NA	NA	NA
33	Control	M	70	NA	NA	NA	NA
30	Control	M	57	NA	NA	NA	NA
18	Lung	F	79	77	4	Y	3 weeks—6 months
22	Lung	M	75	66	1A	Y	>1 year
9	Lung	M	65	63	2B	Y	3 weeks—6 months
19	Lung	F	76	64	4	Y	>1 year
13	Lung	F	70	68	4	N	NA
14	Lung	F	58	56	4B	Y	3 weeks—6 months
24	Lung	F	69	67	4B	Y	3 weeks—6 months
2	Lung	M	78	76	4A	N	NA
3	Lung	M	82	80	4A	Y	3 weeks—6 months
4	Lung	F	62	60	4B	N	NA
8	Lung	F	63	61	4B	N	NA
11	Lung	F	68	66	4A	Y	3 weeks—6 months
6	Lung	F	76	73	4A	Y	3 weeks—6 months
15	Lung	F	71	69	4A	N	NA
25	Prostate	M	64	55	2B	Y	>1 year
16	Prostate	M	71	64	4	Y	>1 year
12	Prostate	M	79	68	4	Y	>1 year
23	Prostate	M	59	57	4	Y	3 weeks—6 months
5	Prostate	M	74	70	4B	Y	>1 year
7	Prostate	M	80	61	4	Y	>1 year
31	Prostate	M	80	69	4	Y	>1 year
28	Prostate	M	67	55	4	Y	>1 year
29	Prostate	M	88	81	4	Y	>1 year
21	Prostate	M	78	64	1	Y	>1 year

## Data Availability

Data used to generate these results have been uploaded as Appendix A. Raw data may be obtained by contacting either corresponding authors.

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
