# Peer review of "Detectable Lipidomes and Metabolomes by Different Plasma Exosome Isolation Methods in Healthy Controls and Patients with Advanced Prostate and Lung Cancer"

_ijms, 2023, doi:10.3390/ijms24031830_

Round 1

Reviewer 1 Report

The authors investigated three plasma exosome isolation methods on lipidomes and metabolomes profile of healthy controls and patients with advanced prostate and lung cancer. It’s very practical; however, some minor correction should be done in the revised version.

Abstract:

The authors used three extraction methods for the enrichment of exosome in plasma from patients with CRPC and stage IV NSCLC and healthy individuals. The authors should focus on the size exclusion, lectin binding, and TIM4 binding, not SBI, Takara, and Wako. So, I suggest that the authors do not mention the information of producers in the abstract, as well as the results and discussion. What’s the meaning of “TIM4”? The full name should be given when the abbreviation appeared for the first time in the manuscript. The authors used different font in the abstract, please check.

Materials and Methods:

A space should be inserted between number and unit, such as 1.5 mL, not 1.5mL, please check the whole manuscript

Mass Spectrometry Sample Preparation: please provide more detailed information about the lipid extraction

Mass Spectrometry Analysis: no Mass Spectrometry condition has been reported, such as scanning mode, voltage, temperature, scanning range, etc.

Results:

The authors used different font in the text, please check the whole manuscript.

Reviewer 2 Report

The work entitled "Detectable lipidomes and metabolomes by different plasma exosome isolation methods in healthy controls and patients with advanced prostate and lung cancer" provides insights into the plausible biomarker search into the exosomal fraction for cancer detection/diagnosis. The manuscript brought interesting comparisons between 3 methods for exosomal isolation, 2 based on exosomal protein binding and 1 on particle size exclusion. The samples used involve blood from healthy donors as controls and blood samples from prostate and lung cancer patients. Although this manuscript involves an enormous workload in cutting-edge metabolomics/lipidomics techniques, no substantial conclusion is achieved about a profile of metabolites/lipid contents that certainly allows the diagnosis of one of the analyzed cancer types. Therefore, although the paper brings information that can be useful in the future as guidance for biomarker investigation, the manuscript just presents a descriptive analysis of the isolated metabolites/lipids and a comparison of the 3 exosomal isolation methods. Indeed the conclusion of the manuscript is that the size exclusion (SBI) method is: "preferred to identify cancer-derived biomarkers...". Based on the aforementioned evaluation, I consider that the evidence presented is preliminary and should be complemented with other techniques/analysis and maybe with more samples from other cancer types patients in order to reach a conclusion about cancer biomarkers that can be published in IJMS.

Author Response

We would like to thank the reviewer for their time and consideration of the manuscript.

Round 2

Reviewer 1 Report

Accept in present form

Reviewer 2 Report

Reply to the authors (2nd review):

I understand that the time and money applied to this work were not rewarded by the results (expected as a prospective biomarker or set o distinctive features -biomarkers- that can identify cancer patients in a population or at least a hint of exosomal lipid composition that can be associated with... ). Nevertheless, the information the article brings could be valuable for other groups working on this topic and pharmaceutical companies searching for distinctive features for early diagnosis. In this sense, negative data along with new information from OMICS should be also regarded as valuable publishable information. In this regard, Scientific Journals should review the publication policies, in order to not only incorporate “cutting edge hypothesis-driven discoveries” that bring new “Knowledge”, but also consider for publication descriptive works that bring negative data and/or just “information”. Having said this, upon the incorporation of a final suggestion, I delegate the definitive decision to publish the present manuscript to the editors (who can better understand the scope of IJMS and the suitability of the present study for its inclusion in the next issue).

Final suggestion:

Replace in the Conclusion section the sentence:

“Our study evaluated the lipidome and metabolome profiling of plasma exosomes enriched with three commercially available exosome extraction methods”

For a more realistic:

“In this study, we set out to identify which commercially available plasma exosome isolation kit, …... provided the most information that could be used in comparing patient groups” (a phrase that the authors suggested) or a similar phrase depicting the true goal of the present paper.